# Add-On Anomaly Threshold Technique for Improving Unsupervised Intrusion Detection on SCADA Data

**Abdulmohsen Almalawi [1], Adil Fahad [2,*], Zahir Tari [3], Asif Irshad Khan [1],**
**Nouf Alzahrani [2,*], Sheikh Tahir Bakhsh [1], Madini O. Alassafi [1],**
**Abdulrahman Alshdadi [4] and Sana Qaiyum [5]**

[1] School of Computer Science & Information Technology, King Abdulaziz University, Jeddah 21589,
Saudi Arabia; balmalowy@kau.edu.sa (A.A.); aikhan@kau.edu.sa (A.I.K.); stbakhsh@kau.edu.sa (S.T.B.);
malasafi@kau.edu.sa (M.O.A.)

[2] Department of Computer Science, College of Computer Science & Information Technology,
Al Baha University, Al Baha 65527, Saudi Arabia

[3] Distributed Systems and Networking (DSN) Discipline, School of Computer Science and Information
Technology (CSIT), RMIT University, Melbourne, VIC 3000, Australia; zahir.tari@rmit.edu.au

[4] Department of Information Systems and Technology, College of Computer Science & Engineering,
Jeddah University, Jeddah 23218, Saudi Arabia; alshdadi@uj.edu.sa

[5] Center for Research in Data Sciences, Universiti Teknologi PETRONAS, Seri Iskandar 32610, Malaysia;
sqaiyum.cs@gmail.com

* Correspondence: afalharthi@bu.edu.sa (A.F.); Noufalzahrani@bu.edu.sa (N.A.); Tel.: +966-539392235 (A.F.)

**Abstract:** Supervisory control and data acquisition (SCADA) systems monitor and supervise our daily infrastructure systems and industrial processes. Hence, the security of the information systems of critical infrastructures cannot be overstated. The effectiveness of unsupervised anomaly detection approaches is sensitive to parameter choices, especially when the boundaries between normal and abnormal behaviours are not clearly distinguishable. Therefore, the current approach in detecting anomaly for SCADA is based on the assumptions by which anomalies are defined; these assumptions are controlled by a parameter choice. This paper proposes an add-on anomaly threshold technique to identify the observations whose anomaly scores are extreme and significantly deviate from others, and then such observations are assumed to be "abnormal". The observations whose anomaly scores are significantly distant from "abnormal" ones will be assumed as "normal". Then, the ensemble-based supervised learning is proposed to find a global and efficient anomaly threshold using the information of both "normal"/"abnormal" behaviours. The proposed technique can be used for any unsupervised anomaly detection approach to mitigate the sensitivity of such parameters and improve the performance of the SCADA unsupervised anomaly detection approaches. Experimental results confirm that the proposed technique achieved a significant improvement compared to the state-of-the-art of two unsupervised anomaly detection algorithms.

**Keywords:** SCADA security; intrusion detection; unsupervised learning; Industrial Internet of Things (IIoT); information-security; security threats; vulnerability measurement

---

## 1. Introduction

Supervisory control and data acquisition (SCADA) systems control and monitor the information systems of industrial and critical infrastructure (such as electricity, gas, and water). Recently, there has been an increase in attacks targeting these systems. Compromising the information systems of SCADA

can lead to large financial losses and serious impacts on public safety and the environment. The attack on a sewage treatment system in Maroochy Shire, Queensland, is an obvious example of the seriousness of cyber attacks on critical infrastructure [1]. Therefore, securing and protecting these systems is *extremely* important [2–4]. The new generation of intrusion detection systems (IDSs) will need to detect ad-hoc SCADA-specific attacks that cannot be detected by existing security technologies [5]. Due to the differences between the nature and the characteristics of traditional IT and SCADA systems, there is a need for the development of SCADA-specific IDSs, and in recent years this has become an interesting research area [6–8].

A SCADA system monitors and controls a series of process parameters. The values of these supervised parameters can reflect the internal representation of the SCADA system. The values are called "SCADA data" which are found in [6,8–13] to be a good information source to monitor the internal behaviour of the given system and protect it from malicious actions that are intended to sabotage or disturb the proper functionality of the targeted system. Therefore, the monitoring of the behaviour of SCADA systems through the evolution of SCADA data has attracted the attention of researchers. In [9,10], a predefined threshold (e.g, minimum, maximum) is proposed to monitor each process parameter individually, and any reading that is not inside a prescribed threshold is considered as an anomaly. These approaches are good for monitoring one single process parameter. However, the value of an individual process parameter may not be abnormal, but in combination with other process parameters, may produce an abnormal observation, which very rarely occurs. These types of parameters are called multivariate parameters, and are assumed to be directly (or indirectly) "correlated".

In [11], the authors proposed an analytical approach to manually identify the range of critical states for multivariate process parameters, and the identified ranges are then used to monitor the critical state of the analysed process parameters. However, analytical approaches require expert involvement, and this results in time-intensive processing that is prone to human error. To avoid the aforementioned issues, purely "normal" SCADA data are used to model the normal behaviours. For example, Rrushi [12] applied probabilistic models to estimate the normalcy of the evolution of multivariate process parameters. Zhanwei et al. [6] proposed a combination of a normal control behaviour model and a normal process behaviour model to build SCADA data-driven detection models to monitor abnormal behaviours in industrial equipment. Gao et al. [8] proposed a neural network-based model to learn the normal behaviours for water tank control systems. Similarly, Zaher et al. [13] proposed the same technique to build the normal behaviours for a wind turbine to identify faults or unexpected behaviours (anomalies).

However, it is difficult to build the "normal" behaviours of a given system using observations of the raw SCADA data because, firstly, it cannot be guaranteed that all observations represent one behaviour as either "normal" or "abnormal", and therefore domain experts are required for the labelling of each observation, and this process is prohibitively expensive; secondly, in order to obtain purely "normal" observations that comprehensively represent "normal" behaviours, this requires a given system to be run for a long period under normal conditions, and this is not practical; and finally, it is challenging to obtain observations that will cover all possible abnormal behaviours that can occur in the future. Therefore, we strongly believe that the development of a SCADA-specific IDS that uses SCADA data and operates in unsupervised mode, where the labelled data is not available, has great potential as a means of addressing the aforementioned issues.

The unsupervised IDS can be a time and cost-efficient means of building detection models from unlabelled data; however, this requires an efficient and accurate technique to differentiate between the normal and abnormal observations without the involvement of experts, which is costly and prone to human error. Then, from observations of each behaviour, either normal or abnormal, the detection models can be built. Two assumptions must be made in unsupervised anomaly detection approaches [14–16]: (i) the number of normal observations in the dataset vastly outperforms the abnormal observations, and (ii) the abnormal observations must be statistically different from

normal ones. Therefore, the performance of the proposed detection models relies mainly on the two aforementioned to distinguish between normal and abnormal behaviours. The reporting of anomalies in the unsupervised mode can be done either by scoring-based or binary-based techniques [17].

Anomaly scoring techniques are categorised into two broad classes: the distance-based and the density-based scoring techniques. The basic idea of the distance-based technique is to distinguish an observation as outlier on the base of the distance to its nearest neighbours, while in the density-based one, an observation is considered as outlier when it lies in a low-density area of its nearest neighbours [17]. All observations in a dataset are given an anomaly score, and therefore actual anomalies are assumed to have the highest scores. The key problem is how to find the best cut-off threshold that minimises the false positive rate while maximising the detection rate. On the one hand, binary-based techniques [14] group similar observations together into a number of clusters. Abnormal observations are identified by making use of the fact that abnormal observations will be considered as outliers, and therefore will not be assigned to any cluster, or they will be grouped into small clusters that have some characteristics which are different from normal clusters. However, labelling an observation as an outlier or a cluster as anomalous is controlled through some parameter choices within each detection technique. For instance, given the top 50% of the observations which have the highest anomaly scores, these are assumed as outliers. In this case, both detection and false positive rates will be higher. Similarly, labelling a low percentage of largest clusters as normal in clustering-based intrusion detection techniques, will result in higher detection and false positive rates. Therefore, the effectiveness of unsupervised intrusion approaches is sensitive to parameter choices, especially when the boundaries between normal and abnormal behaviours are not clearly distinguishable.

This paper proposes the global anomaly threshold to unsupervised detection (GATUD) that is based on anomaly density-based technique because it is considered as one of the anomaly scoring techniques in network anomaly detection [18]. The proposed GATUD approach can be used as an add-on threshold technique to allow unsupervised anomaly scoring-based techniques to set the value of the cut-off threshold parameter at a satisfactory level to guarantee a high detection rate, while minimising the resulting high false positive rate. In addition, it can be used as a robust technique for labelling clusters to improve the accuracy of clustering-based intrusion detection systems. Figure 1 shows that GATUD involves two steps: (i) establishing two small most-representative datasets, where each dataset represents one-class problem (normal or abnormal) with high-confidence; and (ii) using the established datasets to build an ensemble-based decision-making model using a set of supervised classifiers.

This paper is organised as follows. Section 2 presents an overview of related work. Section 3 introduces GATUD. Section 4 presents the experimental set-up, followed by results and discussion in Section 5. Section 6 concludes the paper.

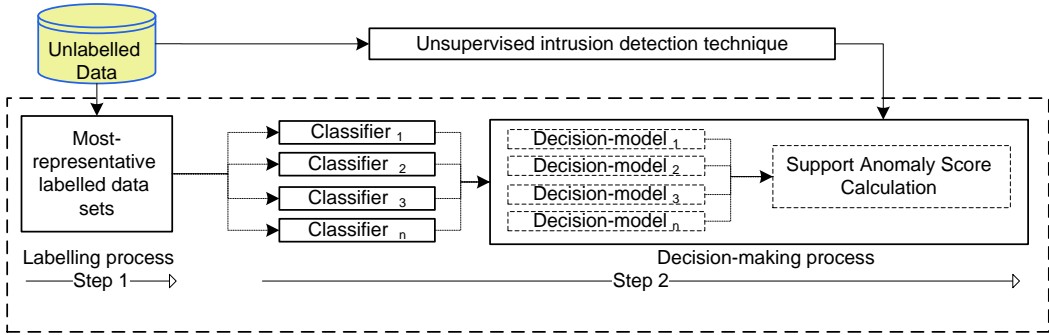

**Figure 1.** Overview of GATUD.

## 2. Related Work

An intrusion detection system (IDS) is a main component in securing computer systems and networks. In the case of SCADA systems, a number of tailored IDSs have been proposed (refer to a survey paper [19]). There are two categories of IDSs: signature-based and anomaly-based. The former detects only known attacks because it monitors the system against specific attack patterns. The latter attempts to build models from the normal behaviour of the systems, and any deviation from this behaviour is assumed to be a malicious activity. Both approaches have advantages and disadvantages. The former achieves good accuracy, but fails to detect attacks that are new or the patterns of which are not learned. Although the latter is able to detect novel attacks, the overall detection accuracy of this approach is low.

This paper focuses on the anomaly detection techniques since they are able to address the problem of the zero-day attacks. Rrushi [12] applied statistics and probability theory to estimate the normality of the evolution of values of correlated process parameters. In the work of [20], the authors assumed that communication patterns among SCADA components are well-behaved, and combined the normal behaviour of SCADA network traffic with artificial intrusion observations to learn the boundaries of the normal behaviour using the neural network technique. There are two types of anomaly detection techniques: supervised and unsupervised modes. In the former mode, training data are labelled, while in the latter, data are not labelled. In contrast to conventional information technology (IT), the unsupervised mode has not been used much in SCADA systems. This is because SCADA security research is relatively new compared with IT. In addition, the security requirements of such systems require a high detection accuracy which this mode lacks.

Recently, machine learning techniques have been successfully applied in unsupervised IDS for industrial control systems [21,22]. Bhatia et al [21] proposed an unsupervised model that uses deep learning autoencoders based on artificial neural networks not for dimension reduction only, but for classification to learn the benign network traffic. The key contribution of their proposed model is to identify the minimal latent subspace that contains the essential characteristics of the benign traffic. Similarly, in [22], the authors proposed unsupervised model that utilises the sparse and denoising auto-encoder (DAE) to obtain the robust latent representations by introducing a stochastic noise to the original data. In [23–25], unsupervised anomaly scoring approaches based on clustering techniques were proposed to detect normal and abnormal behaviours of industrial control systems at network and environmental parameters levels. Each observation is given an anomaly score, and therefore actual anomalies are assumed to have the highest scores. However, the key problem is how to find the near-optimal cut-off threshold that minimises the false positive rate while maximising the detection rate. In order to overcome this issue, this paper proposes an approach inspired by the work proposed in [26]. The authors assign an anomaly-scoring score for each observation in the unlabelled data; the observations that have the highest anomaly scores are labelled as outliers, while the rest are labelled as normal. They randomly selected a subset of normal data and combined it with outliers to create labelled data. Afterwards, a supervised technique was trained with the labelled data to build an outlier filtering rule that differentiates outliers from normal data. However, our approach differs in that we learn the labelled data from data about which we have no prior knowledge, and a set of supervised classifiers are used to build a robust decision-maker because each classifier can capture different knowledge [27]. Finally, our approach is proposed as an add-on component (not an independent technique) for unsupervised learning algorithms in order to benefit from the inherent characteristics of each algorithm.

## 3. The Proposed GATUD Approach

We focus on improving the detection accuracy of unsupervised anomaly detection approaches. This is because such approaches are able to detect (unknown) zero-day attacks. However, they suffer from low accuracy. In this section, we present GATUD that is intended to address this problem. We outline the various steps below.

### 3.1. Learning of Most-Representative Datasets

In this step, two small, most-representative datasets are established from the unlabelled data, where the first and second datasets approximately represent the normal and abnormal behaviours respectively. In order to choose the most-representative datasets, two steps are followed:

#### 3.1.1. Step 1: Anomaly Scoring

Since we have no prior knowledge about the normal and abnormal data, the $k$-nearest neighbour notion is adapted to assign an anomaly score to each observation. The $k$-nearest neighbour notion is chosen because it has produced significant results in anomaly scoring as proposed in [25], in cases where normal data in $n$-dimension space form dense areas, and the abnormal data are sparsely distributed. Unlike the previous approaches, we are concerned with the most relevant normal and abnormal data rather than with the detection of all anomalies. However, the $k$-nearest neighbour algorithm is computationally expensive and this issue has been addressed by the proposed the $k$NNVWC approach in [28], and therefore, it is used to efficiently find $k$-nearest neighbours for each observation in a dataset. Let $S$ be unlabelled dataset of SCADA data with a multi-dimensional space, $m \times n$ matrix, where $m$ and $n$ represent the numbers of observations and attributes in $S$ respectively. Each dimension represents a distinct data point (e.g., temperature, motor speed or humidity), while each observation $x_i$ is represented by values of a set of attributes $A = \{a_1, a_2, a_3, \ldots, a_n\}$. Let $d$ be the Euclidean distance between two observations $x_1 = \{x_{1,1}, x_{1,2}, \ldots, x_{1,n}\}$ and $x_2 = \{x_{2,1}, x_{2,2}, \ldots, x_{2,n}\}$,

$$d(x_1, x_2) = \sqrt{\sum_{i=1}^{n}(x_{1,i} - x_{2,i})^2} \tag{1}$$

where $n$ is the number of the attributes. Given $B = \{b_1, b_2, \ldots, b_k\}$ be a set of $k$-nearest neighbours of the observation $x_i$ where $B \subset S$, $x_i \in S$, and $x_i \notin B$. Then the anomaly score of $x_i$ is computed as follows:

$$\rho(x_i, B) = \frac{1}{k}\sum_{j=1}^{k} d(x_i, b_j) \tag{2}$$

The fast $k$-nearest neighbour algorithm in [28] is used here to find $k$-nearest neighbours. Algorithm 1 summarises the calculation steps of anomaly scores for each an observation $x_i$.

---

**Algorithm 1:** Anomaly scoring calculation.

**Input:** S
/* A matrix of unlabelled SCADA measurement data consisting of $m$ observations and $n$ attributes                                        */
**Input:** k
/* A positive integer that specifies the number of nearest neighbours    */
**Output:** AnomalyScoresList
/* list of Anomaly Scores sorted by rank in descending order            */
1  $AnomalyScoresList \longleftarrow \varnothing$;
2  **foreach** $x_i$ *in S* **do**
3      $\quad B \longleftarrow \text{knn}(x_i, k)$;
        $\quad$/* $k$-nearest neighbours of the observation $x_i$          */
4      $\quad Score \longleftarrow \rho(x_i, B)$;
        $\quad$/* Compute anomaly score as Equation (2)                    */
5      $\quad$put $Score$ in $AnomalyScoresList$;
6  **return** [AnomalyScoresList];

---

3.1.2. Step 2: Selection of Candidate Sets

From the list of anomaly scores, which are produced by Algorithm 1, we establish two small, most-representative datasets, where each dataset represents normal or abnormal behaviours with high confidence. Based on the two previously-mentioned assumptions of normal and abnormal behaviours in the unsupervised mode, we group the list of anomaly scores into three categories, as illustrated in Figure 2: *confidence area of anomalies*, *uncertain area*, and *confidence area of normality*. As shown in this figure, the extent of these areas is determined by the *confidence* thresholds $\beta$, $\alpha$, and $\lambda$. For instance, the smaller threshold $\beta$ of the confidence area of anomalies, the greater the confidence that the observations falling into this area are abnormal. This is true for the confidence area of the normality. Therefore, the thresholds ($\beta$ and $\lambda$) should be kept at a distance from the uncertain area because this area requires a best cut-off threshold in order to judge an observation as either normal or abnormal, especially when some actual anomalous observations have anomaly scores that are close to some normal ones. Therefore, the most-representative datasets for normal and abnormal behaviours are established from the following two categories: *confidence area of normality* and *confidence area of anomalies* respectively. The most-relevant anomalies, *AbnormalData*, are defined by selecting observations whose indices correspond to the top $n$ of *AnomalyScoresList*, where $n = \beta \times |AnomalyScoresList|$. The most relevant normal observations, *MostNormal*, are defined by selecting observations whose indices correspond to the bottom $n$ of *AnomalyScoresList*, where $n = \lambda \times |AnomalyScoresList|$.

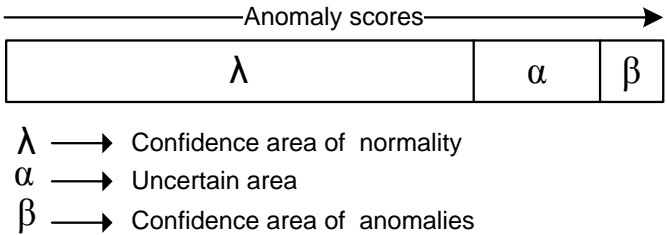

**Figure 2.** The categorization of unlabelled data after applying the anomaly-scoring technique.

Again, if the two assumptions [14] about the unlabelled data are met, the thresholds $\beta$ and $\lambda$ are not difficult to determine. According to these assumptions, the anomalies are assumed to constitute a small portion of the data, where this percentage is assumed to not exceed 5%. Since we are not interested in finding all anomalies more than finding the fraction of anomalies with high confidence, and also we are not supposed to approach the uncertain area, the value of $\beta$ will be set to a value that is smaller than 5%. As opposed to anomalies, the normal data is assumed to constitute a large portion of the data; therefore, setting $\lambda$ to a small value will result in a small dataset of the most-relevant normal observations. However, this dataset might not approximately represent the large portion of the normal data. To address this problem, we propose to set $\lambda$ to a large value providing this value does overlap with the uncertain area by, for example, 80%. This will result in a large dataset that is the most-relevant normal. However, the computation time in the ensemble-based decision-making model will be substantially higher.

In order to resolve the previous problem, we extract a small set of representative observations from the most relevant normal dataset. In this step, we group the similar observations together in terms of Euclidean distance, and take their mean as a representative observation for each group. $k$-means clustering technique [29] is a candidate algorithm for this process because of its simplicity, low computation time, and fast convergence. Moreover, the main disadvantages of $k$-means of determining the appropriate number of clusters and forcing an outlier observation to be assigned to the closest cluster, even if it is dissimilar to its members, will not be problematic in this step. This is because we are not interested in finding specific clusters more than chopping the data into a number of groups, and also the clustering data (the most-relevant normal dataset) are assumed to be outlier-free.

Therefore, the number of clusters $k$ will be set to a small value, where $k \ll |D|$. Algorithm 2 summarises the steps involved in learning a small representative dataset of the most relevant normal observations.

The two small datasets (the learned normal and anomalous datasets) are combined in order to form a labelled compressed representation of the unlabelled data. The concept of this compressed dataset is slightly similar to the concept of the set of support vectors built by a support vector machine (SVM) [30].

---

**Algorithm 2:** Learning a small, representative dataset of the most-relevant normal observations.

---

**Input:** k
```
/* A positive integer that specifies the number of clusters         */
```
**Input:** NormalData
```
/* The observations that fall into the confidence area of normality  */
```
**Output:** RepNormalData
```
/* small dataset that contains representative observations for the
   most-relevant normal observations                               */
```
1  Initialise the cluster centroids $C = \{c_1, c_2, \ldots, c_k\}$ ;

2  *AssignmentList* $\longleftarrow \varnothing$;
```
/* list of tuples < observation, Cluster ID >                        */
```
3  **while** *termination criterion is not met* **do**

4      **foreach** *observation in NormalData* **do**

5         $ClusterID \longleftarrow$ `ClosestCluster`(*observation, C*);
```
          /* Find the closest cluster to this observation          */
```
6         put $< observation, ClusterID >$ in *AssignmentList*;

7      $C \longleftarrow$ `UpdateCentroids`(*AssignmentList*);

8  *RepNormalData* $\longleftarrow C$;
```
/* the centroids of clusters are used as representative observations for
   most-relevant normal observations                               */
```
9  **return** $[RepNormalData]]$;

---

### 3.2. Decision-Making Model

This section introduces the *ensemble-based decision-making model* (EDMM) used to calculate the support anomaly score for each testing observation. As shown in Figure 1, EDMM is composed of a set of supervised classifiers whose individual decisions are combined to form an ensemble decision. This is because the combining of classifiers promised to be effective [31]. Each classifier $c_i$ is trained with the labelled dataset to build a decision model $m_i$. In GATUD, the number and the type of involved supervised classifiers have been left open because the choice of a specific algorithm is a critical step.

Let $C = \{c_i | 1 \leq i \geq n\}$ be a set of candidate supervised classifiers that build a set of decision models $M = \{m_i | 1 \leq i \geq n\}$. Each decision model $m_i$ assigns binary-decision value (either "1" or "0") to a testing observation $x_i$, $m_i(x_i) : v_i$. When the binary value $v_i$ is "1", the observation $x_i$ is judged as anomalous, and otherwise is judged as normal. Then the calculation of the support anomaly score is defined as follows:

$$support(x_i) = \frac{\sum_{j=1}^{n} m_j(x_i)}{n} \tag{3}$$

where $n$ is the number of decision models involved in the calculation of the support anomaly score. The observation type (class), whether abnormal or normal, is defined by the following equation:

$$Class(x_i) = \begin{cases} support(x_i) \geq \tau & \text{Abnormal} = 1 \\ Otherwise & \text{Normal} = 0 \end{cases} \tag{4}$$

where $\tau$ is the percentage of the accepted vote of the decision-models to judge a testing observation $x_i$ as anomalous. For instance, when the threshold $\tau$ is set to 1, the testing observation will not be considered as anomalous unless all involved decision models agree that the observation is an anomaly.

Illustrative Example

A simple example is given below to illustrate the process of the EDMM. Given five supervised classifiers are selected, $C = \{c_1, c_2, c_3, c_4, c_5\}$, and trained with labelled datasets that have been learned at Section 3.1.2, to build the decision models, $M = \{m_1, m_2, m_3, m_4, m_5\}$, and given a testing observation $x_i$ whose status predicted by these in question models as shown in Table 1, then the support anomaly score is computed as follows:

$$support(x_i) = \frac{1+1+0+1+1}{5} = \frac{4}{5} = 0.80$$

Given the threshold $\tau$ set to 0.6, where the observation $x_i$ is considered as an anomaly, at least three decision models have to assign it as anomalous. Therefore, from this example, the observation $x_i$ will be considered anomalous.

**Table 1.** Prediction results for a set of decision models on a testing observation $x_i$.

| Decision Model | $m_1$ | $m_2$ | $m_3$ | $m_4$ | $m_5$ | Sum |
|---|---|---|---|---|---|---|
| Is observation $x_i$ anomalous? | 1 | 1 | 0 | 1 | 1 | 4 |

## 4. The Experimental Setup

To provide quantitative results for GATUD, we use seven labelled datasets, and the normalisation is applied to all these datasets to improve the accuracy and efficiency of GATUD.

### 4.1. Datasets

Five datasets are publicly available [32,33] and two are generated by the proposed SCADA testbed called SCADAVT in [25]. The simulated datasets consist of 12,000 objects, each being described by 113 features. Each feature represents one sensor or actuator reading in the water distribution systems. Each dataset has 100 abnormal observations, where abnormal observations are generated by different attacks. The simulated datasets will be denoted as *SimData*1 and *SimData*2.

The first real dataset comes from the daily measures of sensors in an urban waste water treatment plant (referred to as *DUWWTP*), and it consists of 38 data points (attributes) [33]. This dataset consists of approximately 527 observations, while 14 observations are labelled as abnormal. For more quantitative results, we evaluated our approach on four real datasets that are collected from a real wireless sensor network [32]. Each dataset consists of two attributes (e.g., temperature and humidity). Each dataset has more than 4000 observations, and a tiny portion of abnormal observations. For simplicity, we refer to these datasets as: multi-hop outdoor real data, multi-hop indoor real data, single-hop outdoor real data, and single-hop indoor real data as *MORD*, *MIRD*, *SORD*, and *SIRD*.

### 4.2. Normalisation

To improve the accuracy and efficiency of the proposed approach, the normalisation technique is applied to all testing datasets to scale features by a range 0.0 of 1.0. This will prevent features with a large scale from outweighing features with a small scale. As the actual minimum/maximum of features are already known, and because the identification process is performed in static mode, min-max normalisation technique is used to map the values of features. A given feature $A$ will have values in [0.0, 1.0]. Let us denote by $min_A$ and $max_A$ the minimum value and maximum value of $A$

respectively. Then, to produce the normalised value of $v$ ($v \in A$) using the min-max normalisation method, which we denote as $\hat{v}$, the following formula is used:

$$\hat{v} = \frac{v - min_A}{max_A - min_A} \qquad (5)$$

### 4.3. Choice of Parameters

As discussed in Section 3.1, four parameters are required in order to learn the most representative labelled datasets.

- The $k$-nearest neighbours parameter is the influencing factor for the anomaly scoring technique. However, this value is insensitive and can be heuristically determined based on the assumption that anomalies constitute a tiny portion of the data. Therefore, the value of $k$-nearest is set to be 1% of the representative dataset, because this value is assumed to discriminate between abnormal observations and normal ones in terms of the density-based distance.
- There are three parameters used for learning the most-representative datasets: (i) The extent of confidence area of normality $\lambda$. (ii) The extent of confidence area of abnormity $\beta$. We set the parameters $\lambda$ and $\beta$ to 70% and 1% respectively. Even though the assumption was that the normal and abnormal data constitute larger percentages than the ones we have chosen, we want to maintain some distance from the uncertain area. (iii) The number of clusters $k$ required for $k$-means in the candidate step. The purpose of this parameter is to reduce the number of representative normal observations, not to discover specific clusters. Experimentally, the value of $k$ for several values such as 0.01%, 0.02%, 0.03%, and 0.04% demonstrated similar results, while the larger the value of $k$, the longer the computation time in the anomaly decision-making model of GATUD. Therefore, the value of this parameter is set to be 0.01% of the representative dataset.

### 4.4. The Candidate Classifiers

As previously discussed, the type and the number of the supervised classifiers that are involved in GATUD are left for the implementer. In this paper, a thorough investigation has been conducted of a number of classifiers. We concluded with the most five efficient classifiers. Two are decision-tree based, best-first decision tree (BFTree) [34] and J48 [35]; another two are rule-based, non-nested generalised exemplars (NNge) [36] and projective adaptive resonance theory (PART) [37]; the fifth is a probabilistic based, naive Bayes [38]. When using classifiers, we kept the default parameters of WEKA data mining software [39].

## 5. Results and Discussion

Clearly, GATUD is intended to improve the accuracy of unsupervised anomaly detection systems in general and our proposed SDAD approach [25] in particular. In this evaluation, we demonstrate how GATUD can address the limitations that have been discussed in [25], where a global anomaly threshold is required to work with all datasets that vary in distribution, the number of abnormal observations, and the application domain, when the *scoring-based* technique is adapted. Furthermore, as mentioned earlier, GATUD can be used as an add-on component to help improve the accuracy of the unsupervised anomaly detection approach. We demonstrate its performance when it is used with $k$-means algorithm [29] that is considered as one of the most useful and promising techniques that can be adapted to build an unsupervised clustering-based anomaly detection technique [40,41]. The proposed approach is intended to improve the accuracy of unsupervised intrusion detection systems. Therefore, we evaluate the accuracy of the proposed approach on two types of unsupervised modes: *scoring-based* and *clustering-based* intrusion detection techniques. In this evaluation, the *precision*, *recall*, and *F-measure* metrics are used to quantitatively measure the performance of the proposed approach, because these

metrics are not dependent on the size of the training and testing datasets. The metrics used are defined as follows:

$$Recall = \frac{TP}{TP + FN} \tag{6}$$

$$Precision = \frac{TP}{TP + FP} \tag{7}$$

$$\text{F-measure} = 2 \times \frac{precision \times recall}{precision + recall} \tag{8}$$

where *TP* is the number of anomalies that are correctly detected, *FN* is the number of anomalies that have occurred but not detected, and *FP* is the normal observations that are incorrectly flagged as anomalies. The recall (detection rate) is the proportion of number of anomalies correctly detected to the actual number of anomalies in the testing dataset. The precision metric is used to demonstrate the robustness of the IDS in minimising the false positive rate. However, the system can obtain a high precision score while a number of anomalies are being missed. Similarly, the system can obtain a high recall score, while the false positive rate is higher. Therefore, the F-measure, which is the harmonic mean of precision and recall, would be a more an appropriate metric to demonstrate the accuracy of the proposed approach in this paper and ease the comparison with the other results, because it is the weighted average of the precision and recall rates. Therefore, the F-measure metric takes both false positives and false negatives into account.

### 5.1. Integrating GATUD into SDAD

The separation of the most relevant abnormal observations from normal ones in order to extract proximity detection rules for a given system, is the initial part of the proposed data-driven clustering technique (SDAD) in [25]. However, a cut-off threshold parameter $\eta$ is required to be given, and in fact this parameter plays a major role in separating the most relevant abnormal observations. The demonstrated results were significant; however, various cut-off thresholds $\eta$ for a number of datasets have been used, where some datasets work with a small value of $\eta$, while others work with large values. Therefore, we evaluate how the integration of GATUD into SDAD can help to find a global and efficient anomaly threshold $\eta$ that can work with all datasets, regardless of their variant characteristics, such as distribution, the number of abnormal observations, and the application domain, and meanwhile produces significant results.

It is well-known that the larger the value of the cut-off threshold $\eta$, the higher the detection rate and the higher the false positive rate as well. This, however, will result in poor performance. The determination of an appropriate cut-off threshold $\eta$ that maximises and minimises the detection rate and the false positive rate respectively, is the challenging problem. GATUD addresses this problem by allowing the anomaly scoring technique to choose a large value of cut-off thresholds $\eta$ in order to ensure that the detection rate is higher, while minimising the false positive rate without degrading the detection rate.

In the following, we demonstrate the separation accuracy results with/without the integration of GATUD into SDAD. Then, we demonstrate how this integration has a significant impact on the accuracy of the generation step of proximity detection rules, which is the second phase following the separation process.

#### 5.1.1. The Results of the Separation Process with/without GATUD

Tables 2–8 show the separation accuracy results with/without the integration of GATUD. Clearly, as shown in the result tables, the larger the value of the cut-off threshold $\eta$, the higher the detection rate of abnormal observations. This is because the observations are sorted by their anomaly scores in ascending order and the consideration of the top large portion of the sorted list increases the chance to obtain the actual abnormal observations. However, this will result in a large number of normal observations existing in this portion, and this definitely increases the false positive

rate. On the another hand, it can be seen that the use of GATUD can benefit from the larger value of the cut-off threshold $\eta$ in order to maximise the detection rate of abnormal observations and the obtained list of the assumed abnormal observations is passed through the decision-making model to rejudge whether each observation is abnormal or normal. This, as can be seen in the result tables, nearly sustains the detection rate, and meanwhile minimises the false positive rate.

Table 2 shows that the better accuracy results of separation of abnormal observations without the integration of GATUD are achieved when the cut-off threshold $\eta$ is set to 2.0%, 2.50%, or 3.0%. Moreover, it can be observed that the setting of $\eta$ with larger values, such as 4.5% and 5.0%, has not demonstrated any better results, even though the detection rates of the abnormal observations were high. This is because, as demonstrated, the false positive rates were relatively high. On the other hand, when GATUD was integrated, the high false positive rates were significantly reduced, and meanwhile, the detection rates were sustained at significantly acceptable levels. Similarly, the remaining results for each dataset (as shown in Tables 3–8) demonstrated that the integration of GATUD significantly reduced the high false positive rates when larger values of $\eta$ were used, and meanwhile maintained the detection rates at a satisfactory and significant level. However, the results for the dataset *MORD* in Table 7 were not significant whether GATUD was integrated or not. We would have expected the integration of GATUD to produce significant results, if the cut-off threshold $\eta$ was set to a value that is greater than 0.05%. However, this value is assumed as the maximum percentage of abnormal observations in an unlabelled dataset. Therefore, this dataset is considered to be an exceptional case.

**Table 2.** The separation accuracy of abnormal observations with/without GATUD on DUWWTP.

| | Without GATUD | | | | With GATUD | | | |
|---|---|---|---|---|---|---|---|---|
| $\eta$ | DR | FPR | P | F-M | DR | FPR | P | F-M |
| 0.50% | 14.29% | 0.00% | 100.00% | 25.00% | 14.29% | 0.00% | 100.00% | 25.00% |
| 1.00% | 35.71% | 0.00% | 100.00% | 52.63% | 35.71% | 0.00% | 100.00% | 52.63% |
| 1.50% | 50.00% | 0.00% | 100.00% | 66.67% | 50.00% | 0.00% | 100.00% | 66.67% |
| 2.00% | 70.71% | 0.02% | 99.00% | **82.50%** | 64.29% | 0.02% | 98.90% | 77.92% |
| 2.50% | 79.29% | 0.19% | 92.50% | **85.38%** | 71.43% | 0.13% | 94.34% | 81.30% |
| 3.00% | 87.14% | 0.39% | 87.14% | **87.14%** | 78.57% | 0.13% | 94.83% | 85.94% |
| 3.50% | 92.86% | 0.87% | 76.47% | 83.87% | 92.86% | 0.13% | 95.59% | **94.20%** |
| 4.00% | 92.86% | 1.30% | 68.42% | 78.79% | 92.86% | 0.28% | 90.91% | **91.87%** |
| 4.50% | 94.29% | 1.69% | 62.86% | 75.43% | 92.86% | 0.43% | 86.67% | **89.66%** |
| 5.00% | 100.00% | 2.17% | 58.33% | 73.68% | 92.86% | 0.54% | 83.87% | **88.14%** |

(1) $\eta$: top percentage of observations, which are sorted by anomaly scores in ascending order, are assumed as abnormal. (2) DR: detection rate. (3) FPR: false positive rate. (4) P: precision. (5) F-M: F-measure. (6) Bold value indicates the best F-Measure score.

**Table 3.** The separation accuracy of abnormal observations with/without GATUD on SimData1.

| | Without GATUD | | | | With GATUD | | | |
|---|---|---|---|---|---|---|---|---|
| $\eta$ | DR | FPR | P | F-M | DR | FPR | P | F-M |
| 0.50% | 40.78% | 0.06% | 88.51% | 55.84% | 40.78% | 0.06% | 88.51% | 55.84% |
| 1.00% | 80.78% | 0.13% | 86.74% | 83.65% | 80.78% | 0.13% | 86.74% | 83.65% |
| 1.50% | 98.04% | 0.45% | 70.42% | 81.97% | 98.04% | 0.25% | 80.91% | **88.65%** |
| 2.00% | 98.04% | 0.95% | 52.91% | 68.73% | 98.04% | 0.41% | 72.05% | **83.06%** |
| 2.50% | 98.04% | 1.46% | 42.19% | 59.00% | 98.04% | 0.44% | 70.87% | **82.27%** |
| 3.00% | 98.04% | 1.97% | 35.21% | 51.81% | 98.04% | 0.46% | 69.78% | **81.53%** |
| 3.50% | 98.04% | 2.47% | 30.21% | 46.19% | 98.04% | 0.48% | 68.82% | **80.87%** |
| 4.00% | 98.04% | 2.97% | 26.46% | 41.67% | 98.04% | 0.50% | 68.17% | **80.42%** |
| 4.50% | 98.04% | 3.48% | 23.47% | 37.88% | 98.04% | 0.52% | 67.48% | **79.94%** |
| 5.00% | 98.04% | 3.99% | 21.14% | 34.78% | 98.04% | 0.52% | 67.29% | **79.81%** |

**Table 4.** The separation accuracy of abnormal observations with/without GATUD on SimData2.

| | Without GATUD | | | | With GATUD | | | |
|---|---|---|---|---|---|---|---|---|
| $\eta$ | DR | FPR | P | F-M | DR | FPR | P | F-M |
| 0.50% | 46.50% | 0.01% | 98.94% | 63.27% | 46.50% | 0.01% | 98.94% | 63.27% |
| 1.00% | 81.50% | 0.14% | 85.79% | 83.59% | 81.50% | 0.14% | 85.79% | 83.59% |
| 1.50% | 100.00% | 0.45% | 70.42% | 82.64% | 100.00% | 0.16% | 86.88% | **92.98%** |
| 2.00% | 100.00% | 0.95% | 52.91% | 69.20% | 100.00% | 0.19% | 84.82% | **91.79%** |
| 2.50% | 100.00% | 1.46% | 42.19% | 59.35% | 100.00% | 0.23% | 82.10% | **90.17%** |
| 3.00% | 100.00% | 1.97% | 35.21% | 52.08% | 100.00% | 0.26% | 80.71% | **89.33%** |
| 3.50% | 100.00% | 2.47% | 30.21% | 46.40% | 100.00% | 0.27% | 80.06% | **88.93%** |
| 4.00% | 100.00% | 2.97% | 26.46% | 41.84% | 100.00% | 0.28% | 79.37% | **88.50%** |
| 4.50% | 100.00% | 3.48% | 23.47% | 38.02% | 100.00% | 0.29% | 78.43% | **87.91%** |
| 5.00% | 100.00% | 3.98% | 21.14% | 34.90% | 100.00% | 0.31% | 77.40% | **87.26%** |

**Table 5.** The separation accuracy of abnormal observations with/without GATUD on SIRD.

| | Without GATUD | | | | With GATUD | | | |
|---|---|---|---|---|---|---|---|---|
| $\eta$ | DR | FPR | P | F-M | DR | FPR | P | F-M |
| 0.50% | 17.09% | 0.00% | 100.00% | 29.20% | 17.09% | 0.00% | 100.00% | 29.20% |
| 1.00% | 34.19% | 0.00% | 100.00% | 50.96% | 34.19% | 0.00% | 100.00% | 50.96% |
| 1.50% | 51.28% | 0.00% | 100.00% | 67.80% | 51.28% | 0.00% | 100.00% | 67.80% |
| 2.00% | 68.38% | 0.00% | 100.00% | 81.22% | 68.38% | 0.00% | 100.00% | 81.22% |
| 2.50% | 85.47% | 0.00% | 100.00% | 92.17% | 85.47% | 0.00% | 100.00% | 92.17% |
| 3.00% | 100.00% | 0.08% | 97.50% | 98.73% | 100.00% | 0.08% | 97.50% | 98.73% |
| 3.50% | 100.00% | 0.59% | 83.57% | 91.05% | 100.00% | 0.52% | 85.21% | **92.02%** |
| 4.00% | 100.00% | 1.09% | 73.58% | 84.78% | 100.00% | 0.85% | 78.10% | **87.71%** |
| 4.50% | 100.00% | 1.60% | 65.36% | 79.05% | 100.00% | 0.97% | 75.73% | **86.19%** |
| 5.00% | 100.00% | 2.12% | 58.79% | 74.05% | 100.00% | 0.99% | 75.24% | **85.87%** |

**Table 6.** The separation accuracy of abnormal observations with/without GATUD on SORD.

| | Without GATUD | | | | With GATUD | | | |
|---|---|---|---|---|---|---|---|---|
| $\eta$ | DR | FPR | P | F-M | DR | FPR | P | F-M |
| 0.50% | 71.88% | 0.00% | 100.00% | 83.64% | 71.88% | 0.00% | 100.00% | 83.64% |
| 1.00% | 90.94% | 0.35% | 64.67% | **75.58%** | 90.63% | 0.35% | 64.59% | 75.42% |
| 1.50% | 93.75% | 0.84% | 44.12% | 60.00% | 90.63% | 0.59% | 52.06% | **66.13%** |
| 2.00% | 93.75% | 1.77% | 27.27% | 42.25% | 90.63% | 0.70% | 47.93% | **62.70%** |
| 2.50% | 93.75% | 2.00% | 25.00% | 39.47% | 90.63% | 0.72% | 47.15% | **62.03%** |
| 3.00% | 96.88% | 2.33% | 22.79% | 36.90% | 90.63% | 0.72% | 47.08% | **61.97%** |
| 3.50% | 96.88% | 2.84% | 19.50% | 32.46% | 90.63% | 0.72% | 47.08% | **61.97%** |
| 4.00% | 96.88% | 3.35% | 17.03% | 28.97% | 90.63% | 0.72% | 47.08% | **61.97%** |
| 4.50% | 96.88% | 3.84% | 15.20% | 26.27% | 90.63% | 0.72% | 47.08% | **61.97%** |
| 5.00% | 96.88% | 4.35% | 13.66% | 23.94% | 90.63% | 0.72% | 47.08% | **61.97%** |

**Table 7.** The separation accuracy of abnormal observations with/without GATUD on MORD.

| | Without GATUD | | | | With GATUD | | | |
|---|---|---|---|---|---|---|---|---|
| $\eta$ | DR | FPR | P | F-M | DR | FPR | P | F-M |
| 0.50% | 36.21% | 0.00% | 100.00% | 53.16% | 36.21% | 0.00% | 100.00% | 53.16% |
| 1.00% | 72.41% | 0.00% | 100.00% | 84.00% | 72.41% | 0.00% | 100.00% | 84.00% |
| 1.50% | 86.03% | 0.31% | 79.21% | 82.48% | 75.86% | 0.00% | 100.00% | **86.27%** |
| 2.00% | 86.21% | 0.84% | 58.82% | 69.93% | 75.86% | 0.00% | 100.00% | **86.27%** |
| 2.50% | 86.55% | 1.34% | 47.36% | 61.22% | 75.86% | 0.00% | 100.00% | **86.27%** |
| 3.00% | 87.41% | 1.83% | 39.92% | 54.81% | 75.86% | 0.00% | 100.00% | **86.27%** |
| 3.50% | 87.93% | 2.33% | 34.46% | 49.51% | 75.86% | 0.00% | 100.00% | **86.27%** |
| 4.00% | 87.93% | 2.83% | 30.18% | 44.93% | 75.86% | 0.00% | 100.00% | **86.27%** |
| 4.50% | 88.45% | 3.33% | 27.00% | 41.37% | 75.86% | 0.00% | 100.00% | **86.27%** |
| 5.00% | 89.31% | 3.82% | 24.55% | 38.51% | 75.86% | 0.00% | 100.00% | **86.27%** |

**Table 8.** The separation accuracy of abnormal observations with/without GATUD on MIRD.

| | Without GATUD | | | | With GATUD | | | |
|---|---|---|---|---|---|---|---|---|
| $\eta$ | DR | FPR | P | F-M | DR | FPR | P | F-M |
| 0.50% | 21.00% | 0.00% | 100.00% | 34.71% | 21.00% | 0.00% | 100.00% | 34.71% |
| 1.00% | 42.00% | 0.00% | 100.00% | 59.15% | 42.00% | 0.00% | 100.00% | 59.15% |
| 1.50% | 63.00% | 0.00% | 100.00% | 77.30% | 63.00% | 0.00% | 100.00% | 77.30% |
| 2.00% | 85.00% | 0.00% | 100.00% | 91.89% | 85.00% | 0.00% | 100.00% | 91.89% |
| 2.50% | 100.00% | 0.15% | 94.34% | 97.09% | 97.00% | 0.00% | 100.00% | **98.48%** |
| 3.00% | 100.00% | 0.65% | 78.74% | 88.11% | 97.00% | 0.00% | 100.00% | **98.48%** |
| 3.50% | 100.00% | 1.16% | 67.57% | 80.65% | 97.00% | 0.00% | 100.00% | **98.48%** |
| 4.00% | 100.00% | 1.67% | 59.17% | 74.35% | 97.00% | 0.00% | 100.00% | **98.48%** |
| 4.50% | 100.00% | 2.18% | 52.63% | 68.97% | 97.00% | 0.00% | 100.00% | **98.48%** |
| 5.00% | 100.00% | 2.71% | 47.17% | 64.10% | 97.00% | 0.00% | 100.00% | **98.48%** |

### 5.1.2. The Results of Proximity Detection Rules with/without GATUD

As mentioned previously, the generation process of proximity detection rules comes after and relies on the separation process. Therefore, the robustness of these proximity detection rules is influenced by the accuracy of the separation process, and as shown earlier, the integration of GATUD demonstrated significant results in the separation process even with large cut-off thresholds $\eta$. Therefore, the detection accuracy results of the proximity detection rules, which are extracted from the abnormal and normal observations that were separated using such these large cut-off thresholds $\eta$, are expected to be significant. Tables 9–15 show the detection accuracy results.

Each table represents the results of the detection accuracy results for each individual dataset, and also they are divided into two parts: the first part shows the results of the proximity-detection rules that were extracted by separating abnormal from normal observations where GATUD was not integrated in the separation process. The second part shows the results obtained after the integration of GATUD. The result tables show that the integration of GATUD into the separation process helps to generate robust proximity-detection rules even with large cut-off thresholds $\eta$.

Overall, Table 16 highlights the acceptable thresholds $\eta$ through which the extracted proximity-detection rules demonstrated significant detection accuracy results, where GATUD was integrated into the separation process of abnormal and normal observations. From this table, the determination of the near-optimal value of a cut-off threshold $\eta$ will not be problematic because the value of $\eta$ can be set to 0.05, which is assumed as the maximum percentage of abnormal observations in an unlabelled dataset. The resultant high positive rate that might result from this large value can significantly be reduced by the integration of GATUD.

**Table 9.** The detection accuracy of the proximity-detection rules that have been extracted with/without the integration of GATUD in the separation process on DUWWTP.

| | | Without GATUD | | | | | | With GATUD | | | | | |
|---|---|---|---|---|---|---|---|---|---|---|---|---|---|
| $\eta$ | w | NC | AC | DR | FPR | P | F-M | NC | AC | DR | FPR | P | F-M |
| 0.50% | | 62 | 2 | 14.29% | 0.00% | 100.00% | 25.00% | 62 | 2 | 14.29% | 0.00% | 100.00% | 25.00% |
| 1.00% | | 59 | 5 | 35.71% | 0.00% | 100.00% | 52.63% | 59 | 5 | 35.71% | 0.00% | 100.00% | 52.63% |
| 1.50% | | 57 | 7 | 50.00% | 0.00% | 100.00% | 66.67% | 57 | 7 | 50.00% | 0.00% | 100.00% | 66.67% |
| 2.00% | | 54 | 10 | 70.71% | 0.00% | 100.00% | **82.85%** | 55 | 9 | 64.29% | 0.00% | 100.00% | 78.26% |
| 2.50% | 0.7506 | 52 | 12 | 79.29% | 0.00% | 100.00% | **88.45%** | 54 | 10 | 71.43% | 0.00% | 100.00% | 83.33% |
| 3.00% | | 51 | 14 | 87.14% | 5.85% | 80.26% | 83.56% | 53 | 11 | 78.57% | 0.00% | 100.00% | **88.00%** |
| 3.50% | | 48 | 16 | 92.86% | 0.78% | 97.01% | **94.89%** | 52 | 12 | 84.29% | 0.00% | 100.00% | 91.47% |
| 4.00% | | 46 | 18 | 92.86% | 0.97% | 96.30% | **94.55%** | 52 | 12 | 84.29% | 0.00% | 100.00% | 91.47% |
| 4.50% | | 44 | 20 | 94.29% | 9.75% | 72.53% | 81.99% | 51 | 13 | 85.71% | 0.19% | 99.17% | **91.95%** |
| 5.00% | | 41 | 23 | 100.00% | 11.70% | 70.00% | 82.35% | 49 | 15 | 89.29% | 0.58% | 97.66% | **93.28%** |

(1) $\eta$: top percentage of observations, which are sorted by anomaly scores in ascending order, are assumed as abnormal. (2) w: the cluster width parameter. (3) NC: the number of the produced normal clusters. (4) AC: the number of the produced abnormal clusters. (5) DR: detection rate. (6) FPR: false positive rate. (7) P: precision. (8) F-M: F-measure. (9) Bold value indicates the best F-measure score.

**Table 10.** The detection accuracy of the proximity-detection rules that have been extracted with/without the integration of GATUD in the separation process on SimData1.

| $\eta$ | w | NC | AC | DR | FPR | P | F-M | NC | AC | DR | FPR | P | F-M |
|---|---|---|---|---|---|---|---|---|---|---|---|---|---|
| | | | | **Without GATUD** | | | | | | **With GATUD** | | | |
| 0.50% | | 316 | 39 | 40.78% | 0.06% | 98.58% | 57.70% | 316 | 39 | 40.78% | 0.06% | 98.58% | 57.70% |
| 1.00% | | 286 | 68 | 80.78% | 0.13% | 98.33% | 88.70% | 286 | 68 | 80.78% | 0.13% | 98.33% | 88.70% |
| 1.50% | | 265 | 92 | 98.04% | 0.46% | 95.42% | 96.71% | 272 | 81 | 98.04% | 0.28% | 97.18% | **97.61%** |
| 2.00% | | 252 | 104 | 98.04% | 1.11% | 89.69% | 93.68% | 269 | 82 | 98.04% | 0.42% | 95.79% | **96.90%** |
| 2.50% | 0.1456 | 242 | 116 | 98.04% | 1.73% | 84.75% | 90.91% | 269 | 82 | 98.04% | 0.44% | 95.60% | **96.81%** |
| 3.00% | | 238 | 131 | 98.04% | 2.49% | 79.43% | 87.76% | 268 | 85 | 98.04% | 0.48% | 95.24% | **96.62%** |
| 3.50% | | 234 | 138 | 98.04% | 3.08% | 75.76% | 85.47% | 267 | 85 | 98.04% | 0.56% | 94.52% | **96.25%** |
| 4.00% | | 225 | 144 | 98.04% | 3.64% | 72.52% | 83.37% | 267 | 85 | 98.04% | 0.58% | 94.34% | **96.15%** |
| 4.50% | | 219 | 151 | 98.04% | 4.24% | 69.40% | 81.27% | 266 | 85 | 98.04% | 0.59% | 94.25% | **96.11%** |
| 5.00% | | 211 | 157 | 98.04% | 4.75% | 66.93% | 79.55% | 267 | 85 | 98.04% | 0.59% | 94.25% | **96.11%** |

**Table 11.** The detection accuracy of the proximity-detection rules that have been extracted with/without the integration of GATUD in the separation process on SimData2.

| $\eta$ | w | NC | AC | DR | FPR | P | F-M | NC | AC | DR | FPR | P | F-M |
|---|---|---|---|---|---|---|---|---|---|---|---|---|---|
| | | | | **Without GATUD** | | | | | | **With GATUD** | | | |
| 0.50% | | 307 | 42 | 46.50% | 0.01% | 99.79% | 63.44% | 307 | 42 | 46.50% | 0.01% | 99.79% | 63.44% |
| 1.00% | | 279 | 71 | 70.00% | 0.14% | 97.90% | 81.63% | 279 | 71 | 70.00% | 0.14% | 97.90% | 81.63% |
| 1.50% | | 262 | 90 | 100.00% | 0.51% | 94.97% | 97.42% | 268 | 78 | 100.00% | 0.17% | 98.23% | **99.11%** |
| 2.00% | | 255 | 102 | 100.00% | 1.18% | 89.05% | 94.21% | 268 | 78 | 100.00% | 0.20% | 97.94% | **98.96%** |
| 2.50% | 0.1476 | 243 | 112 | 100.00% | 1.72% | 84.82% | 91.79% | 268 | 78 | 100.00% | 0.26% | 97.37% | **98.67%** |
| 3.00% | | 239 | 120 | 100.00% | 2.42% | 79.87% | 88.81% | 268 | 78 | 100.00% | 0.26% | 97.37% | **98.67%** |
| 3.50% | | 231 | 130 | 100.00% | 3.05% | 75.93% | 86.32% | 268 | 78 | 100.00% | 0.26% | 97.37% | **98.67%** |
| 4.00% | | 228 | 139 | 100.00% | 3.63% | 72.57% | 84.10% | 268 | 78 | 100.00% | 0.28% | 97.18% | **98.57%** |
| 4.50% | | 222 | 146 | 100.00% | 4.19% | 69.64% | 82.10% | 268 | 78 | 100.00% | 0.31% | 96.90% | **98.43%** |
| 5.00% | | 216 | 147 | 100.00% | 4.71% | 67.11% | 80.32% | 268 | 78 | 100.00% | 0.37% | 96.34% | **98.14%** |

**Table 12.** The detection accuracy of the proximity-detection rules that have been extracted with/without the integration of GATUD in the separation process on SIRD.

| $\eta$ | w | NC | AC | DR | FPR | P | F-M | NC | AC | DR | FPR | P | F-M |
|---|---|---|---|---|---|---|---|---|---|---|---|---|---|
| | | | | **Without GATUD** | | | | | | **With GATUD** | | | |
| 0.50% | | 63 | 17 | 17.09% | 0.00% | 100.00% | 29.20% | 63 | 17 | 17.09% | 0.00% | 100.00% | 29.20% |
| 1.00% | | 45 | 35 | 34.19% | 0.00% | 100.00% | 50.96% | 45 | 35 | 34.19% | 0.00% | 100.00% | 50.96% |
| 1.50% | | 36 | 44 | 51.97% | 0.00% | 100.00% | 68.39% | 36 | 44 | 51.97% | 0.00% | 100.00% | 68.39% |
| 2.00% | | 31 | 49 | 67.61% | 0.00% | 100.00% | 80.67% | 31 | 49 | 67.61% | 0.00% | 100.00% | 80.67% |
| 2.50% | 0.0124 | 24 | 57 | 72.65% | 0.00% | 100.00% | **84.16%** | 24 | 57 | 70.09% | 0.00% | 100.00% | 82.41% |
| 3.00% | | 17 | 63 | 100.00% | 0.42% | 98.48% | **99.24%** | 17 | 63 | 100.00% | 0.42% | 98.48% | **99.24%** |
| 3.50% | | 15 | 64 | 100.00% | 0.77% | 97.26% | 98.61% | 16 | 63 | 100.00% | 0.65% | 97.66% | **98.82%** |
| 4.00% | | 15 | 64 | 100.00% | 1.58% | 94.51% | 97.18% | 16 | 63 | 100.00% | 0.84% | 97.01% | **98.48%** |
| 4.50% | | 15 | 64 | 100.00% | 6.98% | 79.59% | 88.64% | 16 | 63 | 100.00% | 0.93% | 96.69% | **98.32%** |
| 5.00% | | 15 | 65 | 100.00% | 8.14% | 76.97% | 86.99% | 16 | 64 | 100.00% | 0.93% | 96.69% | **98.32%** |

**Table 13.** The detection accuracy of the proximity-detection rules that have been extracted with/without the integration of GATUD in the separation process on SORD.

| $\eta$ | w | NC | AC | DR | FPR | P | F-M | NC | AC | DR | FPR | P | F-M |
|---|---|---|---|---|---|---|---|---|---|---|---|---|---|
| | | | | **Without GATUD** | | | | | | **With GATUD** | | | |
| 0.50% | | 73 | 20 | 71.88% | 0.00% | 100.00% | 83.64% | 73 | 20 | 71.88% | 0.00% | 100.00% | 83.64% |
| 1.00% | | 67 | 27 | 90.94% | 0.28% | 95.41% | **93.12%** | 67 | 27 | 90.63% | 0.28% | 95.39% | 92.95% |
| 1.50% | | 64 | 32 | 93.75% | 0.90% | 86.96% | 90.23% | 67 | 28 | 90.63% | 0.60% | 90.63% | **90.63%** |
| 2.00% | | 62 | 33 | 93.75% | 1.32% | 81.97% | 87.46% | 67 | 28 | 90.63% | 0.72% | 88.96% | **89.78%** |
| 2.50% | 0.0152 | 61 | 35 | 93.75% | 2.02% | 74.81% | 83.22% | 67 | 28 | 90.63% | 0.74% | 88.69% | **89.64%** |
| 3.00% | | 57 | 37 | 96.88% | 2.38% | 72.26% | 82.78% | 67 | 28 | 90.63% | 0.74% | 88.69% | **89.64%** |
| 3.50% | | 55 | 41 | 96.88% | 3.03% | 67.10% | 79.28% | 67 | 28 | 90.63% | 0.74% | 88.69% | **89.64%** |
| 4.00% | | 55 | 42 | 96.88% | 3.83% | 61.75% | 75.43% | 67 | 28 | 90.63% | 0.74% | 88.69% | **89.64%** |
| 4.50% | | 54 | 41 | 96.88% | 4.19% | 59.62% | 73.81% | 67 | 28 | 90.63% | 0.74% | 88.69% | **89.64%** |
| 5.00% | | 54 | 42 | 96.88% | 4.59% | 57.41% | 72.09% | 67 | 28 | 90.63% | 0.74% | 88.69% | **89.64%** |

**Table 14.** The detection accuracy of the proximity-detection rules that have been extracted with/without the integration of GATUD on in the separation process MORD.

| | | | | Without GATUD | | | | | | | With GATUD | | |
|---|---|---|---|---|---|---|---|---|---|---|---|---|---|
| $\eta$ | w | NC | AC | DR | FPR | P | F-M | NC | AC | DR | FPR | P | F-M |
| 0.50% | | 74 | 14 | 36.21% | 0.00% | 100.00% | 53.16% | 74 | 14 | 36.21% | 0.00% | 100.00% | 53.16% |
| 1.00% | | 55 | 34 | 72.59% | 0.00% | 100.00% | 84.12% | 55 | 34 | 72.59% | 0.00% | 100.00% | 84.12% |
| 1.50% | | 53 | 38 | 77.59% | 0.56% | 94.54% | 85.23% | 55 | 34 | 77.41% | 0.00% | 100.00% | **87.27%** |
| 2.00% | | 53 | 39 | 86.21% | 1.10% | 90.74% | 88.42% | 55 | 34 | 77.41% | 0.00% | 100.00% | **87.27%** |
| 2.50% | 0.0093 | 52 | 39 | 86.55% | 1.68% | 86.55% | 86.55% | 55 | 34 | 77.41% | 0.00% | 100.00% | **87.27%** |
| 3.00% | | 50 | 40 | 87.41% | 2.20% | 83.25% | 85.28% | 55 | 34 | 77.41% | 0.00% | 100.00% | **87.27%** |
| 3.50% | | 49 | 41 | 87.93% | 2.72% | 80.19% | 83.88% | 55 | 34 | 77.41% | 0.00% | 100.00% | **87.27%** |
| 4.00% | | 48 | 42 | 87.93% | 3.00% | 78.58% | 82.99% | 55 | 34 | 77.41% | 0.00% | 100.00% | **87.27%** |
| 4.50% | | 48 | 43 | 88.45% | 3.39% | 76.57% | 82.08% | 55 | 34 | 77.41% | 0.00% | 100.00% | **87.27%** |
| 5.00% | | 46 | 44 | 89.31% | 3.99% | 73.68% | 80.75% | 55 | 34 | 77.41% | 0.00% | 100.00% | **87.27%** |

**Table 15.** The detection accuracy of the proximity-detection rules that have been extracted with/without the integration of GATUD on in the separation process MIRD.

| | | | | Without GATUD | | | | | | | With GATUD | | |
|---|---|---|---|---|---|---|---|---|---|---|---|---|---|
| $\eta$ | w | NC | AC | DR | FPR | P | F-M | NC | AC | DR | FPR | P | F-M |
| 0.50% | | 62 | 13 | 20.10% | 0.00% | 100.00% | 33.47% | 62 | 13 | 20.10% | 0.00% | 100.00% | 33.47% |
| 1.00% | | 44 | 31 | 42.00% | 0.00% | 100.00% | 59.15% | 44 | 31 | 42.00% | 0.00% | 100.00% | 59.15% |
| 1.50% | | 37 | 38 | 63.00% | 0.00% | 100.00% | 77.30% | 37 | 38 | 63.00% | 0.00% | 100.00% | 77.30% |
| 2.00% | | 26 | 49 | 70.00% | 0.00% | 100.00% | 82.35% | 26 | 49 | 70.00% | 0.00% | 100.00% | 82.35% |
| 2.50% | 0.0135 | 20 | 55 | 100.00% | 0.20% | 99.11% | **99.55%** | 22 | 54 | 96.90% | 0.00% | 100.00% | **98.43%** |
| 3.00% | | 18 | 57 | 100.00% | 0.87% | 96.15% | 98.04% | 22 | 54 | 96.90% | 0.00% | 100.00% | **98.43%** |
| 3.50% | | 18 | 58 | 100.00% | 1.29% | 94.43% | 97.13% | 22 | 54 | 96.90% | 0.00% | 100.00% | **98.43%** |
| 4.00% | | 18 | 59 | 100.00% | 6.54% | 76.92% | 86.96% | 22 | 54 | 96.90% | 0.00% | 100.00% | **98.43%** |
| 4.50% | | 15 | 61 | 100.00% | 7.63% | 74.07% | 85.11% | 22 | 54 | 96.90% | 0.00% | 100.00% | **98.43%** |
| 5.00% | | 16 | 65 | 100.00% | 8.28% | 72.46% | 84.03% | 22 | 54 | 96.90% | 0.00% | 100.00% | **98.43%** |

**Table 16.** The acceptable thresholds $\eta$ that produce better accuracy results of the separation of abnormal observations on each dataset when GATUD was integrated

| | Data Sets | | | | | | | # of Agreement |
|---|---|---|---|---|---|---|---|---|
| $\eta$ | DUWWTP | SimData1 | SimData2 | SIRD | SORD | MORD | MIRD | |
| 0.50% | | | | | | | | 0 |
| 1.00% | | | | | | | | 1 |
| 1.50% | | √ | √ | | √ | √ | | 4 |
| 2.00% | | √ | √ | | √ | √ | | 4 |
| 2.50% | | √ | √ | | √ | √ | √ | 5 |
| 3.00% | √ | √ | √ | √ | √ | √ | √ | 7 |
| 3.50% | | √ | √ | √ | √ | √ | √ | 6 |
| 4.00% | | √ | √ | √ | √ | √ | √ | 6 |
| 4.50% | √ | √ | √ | √ | √ | √ | √ | 7 |
| 5.00% | √ | √ | √ | √ | √ | √ | √ | 7 |

(1) $\eta$: top percentage of observations, which are sorted by anomaly scores in ascending order, are assumed as abnormal. (2) # of agreement: the number of datasets that agree on each separation threshold $\eta$, where the agreement is judged by better accuracy results when GATUD was intergraded.

## 5.2. Integrating GATUD into Clustering-Based Technique

Here we show how GATUD can be integrated not only with the scoring-based intrusion detection technique, but also with the clustering-based technique. The *k*-means algorithm, which is considered as one of the most useful and promising techniques for building an unsupervised clustering-based intrusion detection model [40,41], is chosen to demonstrate the integration effectiveness of GATUD with an unsupervised clustering-based intrusion detection technique. This is because this algorithm already has been adapted by Almalawi et al. [25] to build the unsupervised anomaly detection model to detect abnormal observations, and the results were compared with SDAD [25]. Therefore, it is interesting to demonstrate detection accuracy results with/without the integration of GATUD.

For more details about how this algorithm can be adapted to build an unsupervised anomaly detection model, see [25].

In the adaptation of *k*-means for building an unsupervised anomaly detection model, anomalies are assumed to be grouped in clusters that contain percentages, $\theta$, of the data. Let $C = C_1, C_2, \ldots C_n$ be the sets of clusters that have been created. Then, the anomalous clusters are defined as follows:

$$\acute{C} = \{\acute{C}_1, \acute{C}_2, \ldots \acute{C}_b\} = \sum_{i=1}^{n} |C_i| \leq \theta \tag{9}$$

The remaining clusters $C - \acute{C}$ are labelled as normal. In this evaluation, we assume the real anomalous cluster is the cluster where the majority of its members are actual abnormal observations. Assume that the number of abnormal observations $\geq |\acute{C}_i|/2$. Therefore, the labelling accuracy of the assumed percentage $\theta$ of the data in anomalous clusters is measured by the Labelling Error Rate (LER) for clusters:

$$LER = \frac{\#\ of\ clusters\ incorrectly\ labelled}{\#\ of\ All\ identified\ clusters} \tag{10}$$

When integrating GATUD to label the clusters, the members of each individual cluster pass through the decision-making model to be labelled as either normal or abnormal. Then the cluster is labelled according to the label of the majority of its members. The labelling of clusters by GATUD is given as follows:

$$L(C_i) = \sum_{j=1}^{|C_i|} Class(x_j) \tag{11}$$

where $L(C_i)$ is the number of abnormal observations, which are judged by the decision-making model, in the cluster $C_i$. Then the anomalous clusters are defined as follows:

$$\acute{C} = \{\acute{C}_1, \acute{C}_2, \ldots \acute{C}_b\} = \sum_{i=1}^{n} L(C_i) \geq \varepsilon \times |C_i| \tag{12}$$

where $\varepsilon$ is the percentage of the abnormal observations in a cluster $C_i$ to be labelled as abnormal. In this evaluation, it is set to 0.5.

We evaluate the integration of GATUD as an add-in component with *k*-means, where this component is only used to label the produced clusters as either normal or abnormal. The *k*-means requires two user-specified parameters *k* and $\theta$ to build the unsupervised anomaly detection model from unlabelled data. *k* is the number of clusters, and $\theta$ is the percentage of the data in a cluster to be assumed as malicious. However, the parameter $\theta$ is not required when GATUD is integrated. In this evaluation, we demonstrate the detection accuracy of *k*-means as an independent/dependent algorithm. In the independent use, *k*-means is used to cluster the training dataset and labels each cluster using an assumption of the percentage of the data in a cluster to be assumed as malicious [40,41]. While in the dependent use, GATUD is used as a labelling technique for the produced clusters by *k*-means. The parameters *k* and $\theta$ are set to the same values that have been used in paper [25]. This is because the same datasets are used.

Tables 17–23 show the detection accuracy results of two parts: The first part (without GATUD) shows the detection accuracy results of *k*-means. As shown, the the detection accuracy results of five values of $\theta$ are demonstrated. For example, in the Table 17, the dataset, which was referred to as DUWWTP, was clustered into 50, 60, 70, 80, 90, and 100 clusters using *k*-means algorithm; then, when $\theta$ was set to 0.01 the generated clusters that constitute an overwhelmingly large portion ($\geq$99%) of the training dataset are labelled as normal clusters, and otherwise labelled as malicious ones. As see in the Table 17, the labelling error rate (LER) when 50 clusters generated is 12.40% which means the average percentage of the clusters that were incorrectly labelled in 10 fold cross validation. This is because

their respective labels are not the same as the labels of the majority of their respective members. In the second part (with GATUD), the detection accuracy results of *k*-means when GATUD were integrated to label the clusters instead of the assumption that assumes normal clusters constitute an overwhelmingly large portion, as shown. We show only the results of F-measure, as they are the interesting results to compare. Clearly, the detection accuracy results of *k*-means in detecting abnormal observations were very poor for all datasets when GATUD was not integrated. On the other hand, significant results for some datasets are obtained when GATUD is integrated to label the produced clusters. It is obvious from the results that GATUD can be a promising technique to improve the accuracy of an unsupervised anomaly detection approaches, not only with our SDAD approach proposed in [25], but also, it can be integrated with unsupervised clustering-based anomaly detection models.

**Table 17.** The detection accuracy of *k*-means clustering algorithm with/without GATUD on DUWWTP.

| | Without GATUD | | | | | | | | | | With GATUD | |
| | $\theta = 0.01$ | | $\theta = 0.02$ | | $\theta = 0.03$ | | $\theta = 0.04$ | | $\theta = 0.05$ | | | |
| K | LER | F-M | LER | F-M | LER | F-M | LER | F-M | LER | F-M | LER | F-M |
|---|---|---|---|---|---|---|---|---|---|---|---|---|
| 50 | 12.40% | 69.65% | 29.30% | 59.98% | 43.73% | 50.50% | 53.80% | 44.22% | 60.92% | 39.86% | 6.00% | **79.37%** |
| 60 | 20.17% | 62.36% | 39.08% | 53.36% | 53.11% | 44.37% | 61.92% | 39.03% | 67.63% | 35.62% | 7.17% | **74.14%** |
| 70 | 23.57% | 62.95% | 46.43% | 48.73% | 59.76% | 40.56% | 67.39% | 36.05% | 72.20% | 33.17% | 6.71% | **79.36%** |
| 80 | 33.75% | 57.08% | 54.63% | 44.19% | 66.25% | 37.23% | 72.75% | 33.28% | 76.73% | 30.91% | 5.88% | **79.42%** |
| 90 | 40.44% | 58.62% | 62.17% | 42.73% | 71.96% | 35.87% | 77.17% | 32.26% | 80.27% | 30.10% | 7.11% | **80.28%** |
| 100 | 47.60% | 51.52% | 67.10% | 38.56% | 75.63% | 33.01% | 80.08% | 30.13% | 82.74% | 28.39% | 6.40% | **72.13%** |

(1) LER: labelling error-rate. (2) F-M: F-measure. (3) $\theta$: the percentage of the data in a cluster to be assumed as malicious. (4) K: the number of clusters. (5) Bold value indicates the best F-measure score.

**Table 18.** The detection accuracy of *k*-means clustering algorithm with/without GATUD on SimData1.

| | Without GATUD | | | | | | | | | | With GATUD | |
| | $\theta = 0.01$ | | $\theta = 0.02$ | | $\theta = 0.03$ | | $\theta = 0.04$ | | $\theta = 0.05$ | | | |
| K | LER | F-M | LER | F-M | LER | F-M | LER | F-M | LER | F-M | LER | F-M |
|---|---|---|---|---|---|---|---|---|---|---|---|---|
| 50 | 10.40% | 50.56% | 33.40% | 30.82% | 50.00% | 23.74% | 61.10% | 19.79% | 68.48% | 17.67% | 0.60% | **72.59%** |
| 60 | 18.50% | 37.04% | 42.67% | 24.17% | 58.83% | 19.06% | 68.46% | 15.51% | 74.37% | 14.23% | 0.00% | **68.50%** |
| 70 | 28.86% | 26.97% | 55.29% | 18.66% | 68.81% | 15.02% | 76.00% | 13.32% | 80.51% | 12.44% | 0.29% | **64.13%** |
| 80 | 40.88% | 30.68% | 64.69% | 20.10% | 75.58% | 16.30% | 81.19% | 14.51% | 84.68% | 13.39% | 0.13% | **98.11%** |
| 90 | 41.67% | 24.61% | 67.56% | 17.29% | 77.93% | 14.50% | 83.14% | 12.69% | 86.20% | 11.94% | 0.00% | **86.28%** |
| 100 | 53.90% | 21.15% | 75.05% | 15.34% | 82.90% | 13.22% | 86.88% | 11.96% | 89.28% | 11.35% | 0.40% | **86.77%** |

**Table 19.** The detection accuracy of *k*-means clustering algorithm with/without GATUD on SimData2.

| | Without GATUD | | | | | | | | | | With GATUD | |
| | $\theta = 0.01$ | | $\theta = 0.02$ | | $\theta = 0.03$ | | $\theta = 0.04$ | | $\theta = 0.05$ | | | |
| K | LER | F-M | LER | F-M | LER | F-M | LER | F-M | LER | F-M | LER | F-M |
|---|---|---|---|---|---|---|---|---|---|---|---|---|
| 50 | 10.80% | 74.79% | 31.70% | 48.40% | 48.60% | 36.25% | 59.95% | 29.58% | 67.24% | 25.49% | 0.60% | **99.71%** |
| 60 | 21.00% | 52.54% | 44.58% | 33.94% | 59.89% | 26.06% | 69.17% | 21.77% | 74.80% | 19.17% | 0.33% | **99.61%** |
| 70 | 25.86% | 43.87% | 52.07% | 28.49% | 66.67% | 22.10% | 74.29% | 18.80% | 78.89% | 16.80% | 0.14% | **99.80%** |
| 80 | 35.88% | 33.18% | 61.56% | 22.04% | 73.17% | 17.76% | 79.25% | 15.55% | 82.98% | 14.19% | 0.13% | **99.80%** |
| 90 | 45.56% | 25.24% | 68.28% | 17.73% | 78.00% | 14.83% | 82.97% | 13.33% | 86.04% | 12.42% | 0.56% | **99.22%** |
| 100 | 52.20% | 22.03% | 73.40% | 15.79% | 81.37% | 13.51% | 85.50% | 12.33% | 88.04% | 11.62% | 0.30% | **99.41%** |

**Table 20.** The detection accuracy of *k*-means clustering algorithm with/without GATUD on SIRD.

| | Without GATUD | | | | | | | | | | With GATUD | |
| | $\theta = 0.01$ | | $\theta = 0.02$ | | $\theta = 0.03$ | | $\theta = 0.04$ | | $\theta = 0.05$ | | | |
| K | LER | F-M | LER | F-M | LER | F-M | LER | F-M | LER | F-M | LER | F-M |
|---|---|---|---|---|---|---|---|---|---|---|---|---|
| 50 | 13.20% | 80.09% | 32.30% | 62.74% | 45.87% | 52.44% | 55.35% | 45.76% | 61.68% | 41.40% | 2.80% | **94.09%** |
| 60 | 22.17% | 71.64% | 43.00% | 53.86% | 55.28% | 45.25% | 63.75% | 39.72% | 68.83% | 36.34% | 3.17% | **93.82%** |
| 70 | 27.57% | 64.89% | 49.36% | 48.09% | 60.95% | 40.33% | 68.21% | 35.97% | 72.86% | 33.23% | 2.29% | **94.77%** |
| 80 | 37.50% | 56.39% | 56.06% | 43.16% | 67.04% | 36.51% | 72.88% | 32.95% | 76.70% | 30.72% | 2.38% | **94.43%** |
| 90 | 45.33% | 49.58% | 63.17% | 38.59% | 71.41% | 33.45% | 76.33% | 30.51% | 79.56% | 28.77% | 2.33% | **94.18%** |
| 100 | 52.00% | 42.35% | 68.10% | 34.20% | 75.47% | 30.16% | 79.48% | 28.01% | 81.84% | 26.71% | 2.10% | **94.44%** |

**Table 21.** The detection accuracy of *k*-means clustering algorithm with/without GATUD on SORD.

| | Without GATUD | | | | | | | | | | With GATUD | |
|---|---|---|---|---|---|---|---|---|---|---|---|---|
| | $\theta = 0.01$ | | $\theta = 0.02$ | | $\theta = 0.03$ | | $\theta = 0.04$ | | $\theta = 0.05$ | | | |
| K | LER | F-M | LER | F-M | LER | F-M | LER | F-M | LER | F-M | LER | F-M |
| 50 | 18.00% | 42.53% | 36.40% | 28.45% | 50.27% | 21.83% | 60.60% | 18.01% | 67.52% | 15.64% | 2.00% | **78.90%** |
| 60 | 22.67% | 37.43% | 44.50% | 24.32% | 58.83% | 18.62% | 68.04% | 15.49% | 73.50% | 13.62% | 1.67% | **80.01%** |
| 70 | 33.00% | 26.69% | 54.86% | 17.91% | 66.81% | 14.15% | 73.75% | 12.16% | 78.06% | 10.93% | 2.29% | **79.88%** |
| 80 | 43.00% | 20.41% | 63.69% | 14.05% | 73.58% | 11.52% | 79.16% | 10.17% | 82.53% | 9.34% | 1.50% | **81.96%** |
| 90 | 48.67% | 17.01% | 68.61% | 12.15% | 77.63% | 10.14% | 82.19% | 9.11% | 85.04% | 8.49% | 1.44% | **81.07%** |
| 100 | 58.40% | 14.07% | 74.20% | 10.54% | 81.03% | 9.07% | 84.75% | 8.31% | 87.04% | 7.84% | 1.50% | **83.16%** |

**Table 22.** The detection accuracy of *k*-means clustering algorithm with/without GATUD on MORD.

| | Without GATUD | | | | | | | | | | With GATUD | |
|---|---|---|---|---|---|---|---|---|---|---|---|---|
| | $\theta = 0.01$ | | $\theta = 0.02$ | | $\theta = 0.03$ | | $\theta = 0.04$ | | $\theta = 0.05$ | | | |
| K | LER | F-M | LER | F-M | LER | F-M | LER | F-M | LER | F-M | LER | F-M |
| 50 | 8.20% | 72.61% | 30.80% | 48.03% | 47.07% | 36.94% | 57.00% | 31.08% | 64.12% | 27.23% | 0.20% | **91.53%** |
| 60 | 15.17% | 56.72% | 40.83% | 37.64% | 56.28% | 29.59% | 65.29% | 25.16% | 71.03% | 22.42% | 0.30% | **90.55%** |
| 70 | 24.14% | 44.26% | 48.93% | 30.09% | 62.48% | 24.24% | 70.57% | 20.99% | 75.37% | 19.04% | 0.20% | **91.67%** |
| 80 | 33.13% | 37.11% | 57.19% | 25.44% | 68.79% | 20.90% | 75.22% | 18.52% | 79.13% | 17.05% | 0.10% | **92.84%** |
| 90 | 42.11% | 29.29% | 64.39% | 21.18% | 74.22% | 17.97% | 79.39% | 16.27% | 82.58% | 15.26% | 0.50% | **89.82%** |
| 100 | 49.50% | 24.53% | 69.45% | 18.61% | 77.73% | 16.16% | 81.83% | 14.91% | 84.46% | 14.15% | 0.30% | **91.84%** |

**Table 23.** The detection accuracy of *k*-means clustering algorithm with/without GATUD on MIRD.

| | Without GATUD | | | | | | | | | | With GATUD | |
|---|---|---|---|---|---|---|---|---|---|---|---|---|
| | $\theta = 0.01$ | | $\theta = 0.02$ | | $\theta = 0.03$ | | $\theta = 0.04$ | | $\theta = 0.05$ | | | |
| K | LER | F-M | LER | F-M | LER | F-M | LER | F-M | LER | F-M | LER | F-M |
| 50 | 9.80% | 84.02% | 28.90% | 61.31% | 42.27% | 49.73% | 52.35% | 42.47% | 59.60% | 37.69% | 0.25% | **99.31%** |
| 60 | 14.50% | 75.38% | 36.67% | 53.65% | 51.17% | 43.10% | 60.71% | 36.99% | 66.67% | 33.19% | 0.25% | **99.41%** |
| 70 | 23.57% | 63.42% | 47.71% | 44.59% | 61.00% | 36.16% | 68.61% | 31.70% | 73.17% | 28.97% | 0.25% | **99.51%** |
| 80 | 33.13% | 53.07% | 56.13% | 38.25% | 67.50% | 31.78% | 73.72% | 28.35% | 77.50% | 26.25% | 0.25% | **99.70%** |
| 90 | 40.33% | 45.60% | 61.61% | 33.55% | 71.67% | 28.50% | 77.00% | 25.84% | 80.18% | 24.25% | 0.25% | **99.11%** |
| 100 | 51.90% | 37.75% | 70.35% | 28.72% | 77.97% | 25.19% | 81.75% | 23.38% | 84.10% | 22.28% | 0.25% | **99.70%** |

## 6. Conclusions

This paper proposed an innovative approach, called global anomaly threshold to unsupervised detection (GATUD), which is used as an add-on component to improve the accuracy of unsupervised intrusion detection techniques. This has been done by initially learning two labelled small datasets from the unlabelled data, where each dataset represents either normal or abnormal behaviour. Then, a set of supervised classifiers were trained with question datasets to produce an ensemble-based decision-making model that can be integrated into both unsupervised anomaly scoring and clustering-based intrusion detection approaches. In the former, GATUD is used to mitigate the sensitivity of anomaly threshold, while in the latter, it is used to efficiently label the produced clusters as either normal or abnormal. Experiments show that GATUD demonstrates significant and promising results when it was integrated into a clustering-based intrusion detection approach as a labelling technique for the produced clusters.

**Author Contributions:** A.A. (Abdulmohsen Almalawi) conceived and designed the methodology; A.F. and Z.T. were supervising the project; N.A. and S.T.B. contributed in project admin and developing the software; M.O.A. and A.A. (Abdulrahman Alshdadi) performed the investigation and validation; S.Q. formally analysed the data; A.I.K. wrote the paper. All authors have read and agreed to the published version of the manuscript.

**Funding:** This work was supported by the Deanship of Scientific Research (DSR), King Abdulaziz University, Jeddah, under grant No. (DF-234-611-1441). The authors, therefore, gratefully acknowledge DSR technical and financial support.

**Conflicts of Interest:** The authors declare no conflict of interest.

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
