# Peer review of "Add-On Anomaly Threshold Technique for Improving Unsupervised Intrusion Detection on SCADA Data"

_electronics, doi:10.3390/electronics9061017_

Round 1
Reviewer 1 Report
Summary: This paper proposes a labeling framework for SCADA data using unsupervised learning methods.
Strengths:
- The authors proposed a complete framework to automatically label the training dataset and use the labeled dataset for supervised classifiers to train models for intrusion detection.
- The authors implemented the full pipeline and evaluated their effectiveness.
Weaknesses:
- The novelty is not strong. Basically, the authors used unsupervised learning to create labels. The authors rely on two assumptions they made on the dataset: (1) there are more normal data than abnormal data; (2) statistically, the normal data and abnormal data are different. According to the datasets used by the authors, both of the assumptions seem to be valid. Please provide more fundamental arguments about why these two assumptions should be valid in practice. For example, one of my concerns is that suppose the unlabeled dataset is collected during normal operation of a system, and there is no abnormal data in the dataset. Will this kind of all correct training dataset affect the false positive rate of the generated detection models?
- In the title, the authors state that they are using an adaptive anomaly threshold. Please explain what is really meant by the "adaptive anomaly threshold".
Editorial comments:
Please proofread the paper before submission.
There are a few typos:
- page 2, line 61, "this not" -> "this is not"
- page 4, line 1, "they able" -> "they are able"
- page 4, line 135, "cut-o" -> "cut-off"
- page 5, " a observation" -> "an observation"
Reviewer 2 Report
Please explain Tables 17, 18, 19, 20, 21, 22 and 23.
Reviewer 3 Report
1.The abbreviation should be explained when first using, e.g., SCADA, IDS and etc.
2.The style of figures, tables and etc. should be uniform. The font size in Table 1 is obviously larger than others.
3.The citing of eq. (2) in Algorithm 1 is not correct.
4. In the abstract, it is mentioned that an adaptive anomaly threshold technique is proposed, but I cannot find the adaptive mechanism of the proposed method. It seems that the proposed is a learning method, not an adaptive one.
Reviewer 4 Report
Authors propose a method called Global Anomaly Threshold to Unsupervised
Detection (GATUD) used to improve unsupervised algorithms for the detection of anomalies for SCADA systems.
The approach proposed features some novelty and demonstrates good potentialities in terms of false positives reduction. Furthermore, it does not require a labeled dataset to work, whose creation might be problematic for anomaly detection systems.
However, some concerns should be addressed before the publication of the works.
---------------major concerns------------------------------------The authors clearly demonstrate that the proposed solutions can reduce the number of false positives compared to their previous approaches [33]. However:
- For some datasets used for testing (DUWWTP, SORD, MORD, MIRD) the introduction of GATUD leads to a reduction of the detection rate. Despite that, the final values of the detection rate seem compliant to the requirements fixed by Equation 6. However, no mention was produced about the methodology used to fixed these values. Why did the authors use those values to consider the results "significant"? Is there any other work in the literature supporting this conclusion? Why is it more important to privilege false positives over false negatives?
- The authors compared the proposed method only to their previous work. However, no comparison to other approaches in the literature is provided (e.g. [20], [21]). Authors should compare their results ([33] + GATUD) to other works, if possible.
- No mention is provided about the threshold rho (Equation 4) used to obtain the presented results.
----------------------------------minor concerns---------------------------
- There are several typos and English minor errors (e.g. line 129, 131, 136, 238).
- The use of the verb "say" at line 187 seems too informal and not appropriate to me
- Equation 2: you are using the same index "i" to indicate the observation xi and for the sum. At the same time, you are using D() instead of d(), as indicated in Equation 1. That might be misleading since you used D to indicate the matrix of observations.
Round 2
Reviewer 1 Report
The authors have addressed my concerns. It can be accepted for publication.
Author Response
We thank the reviewer for the motivating comments and for the encouragements
Reviewer 3 Report
Some of mentioned concerns have been addressed, but there still exist some writing problems, e.g., the font format in fig.2. Moreover, the contribution is not very clear and it seems that just using some existing methods to conduct the analysis.
Reviewer 4 Report
According to this reviewer, the authors improved the quality of the work sufficiently to deserve publication.
Author Response
We thank the reviewer for the motivating comments and for the encouragements.
Round 3
Reviewer 3 Report
All comments have been addressed.